# Case Technology in Teaching Professional Foreign Communication to Law Students: Comparative Analysis of Distance and Face-to-Face Learning

Oksana Sheredekina , Irina Karpovich * , Larisa Voronova and Tatyana Krepkaia

Institute of Humanities, Department of Foreign Languages, Peter the Great St. Petersburg Polytechnic University, 195251 Saint Petersburg, Russia
* Correspondence: karpovich.ia@flspbgpu.ru

**Abstract:** Foreign language speaking skills need much practice in order to be improved, which is why it is essential to use different teaching techniques to instruct students effectively. The multiplicity and multitasking of the foreign language teaching process requires the introduction of technology that ensures consistently effective results of professional foreign language learning, i.e., case technology in the algorithmized educational process. The conducted research contributes to the issue of case technology implementation—the algorithm of professional task solutions, aimed at improving students' English-speaking skills—in teaching professional legal English. The aim of the study is to investigate the effectiveness of case technology in the improvement of ESP speaking skills in the course of distance and face-to-face learning. Comparative analysis of the distance and face-to-face format of case technology implementation in teaching professional foreign communication to law students was carried out in the course of the two-semester study conducted at SPbPU (N 120) in the period from January 2020 to July 2021. Mixed qualitative and quantitative methods were applied to collect and analyse data for the study. The results of the study showed that, for such assessment criteria as task completion, discourse competence and meaningfulness, the use of case technology in a face-to-face format was more effective than the one carried out on-line. As it can have some pedagogical implications on the improvement of English-speaking skills while teaching professional legal English at university, additional effort should be taken to ensure the effectiveness of case technology in the course of distance learning.

**Keywords:** algorithmized educational process; case technology; English-speaking skills; legal English; distance learning; face-to-face learning

## 1. Introduction

Globalization has caused serious changes in the life of humanity. It has eliminated barriers between nations, national industries, economics and politics, etc. and opened the world labour market to graduates from all over the world. Consequently, the precedence of speaking skills which ensure the proper intercultural communication has become evident in all spheres of life, including professional ones. They require daily legal support based on the ability of a specialist, i.e., a lawyer, to solve a certain legal problem, not only in their native language, but also in a foreign one. Thus, future lawyers should study professional English at a level that will allow them to carry out professional activities in a foreign language.

Higher legal education involves teaching foreign languages at a professional level. Content analysis of the syllabus for the discipline 'Legal English' shows that, despite the priority of speaking skills, they are formed unsystematically and are mediated by the thematic component of the syllabus. The focus of legal English training is mainly on historical information about the laws of ancient civilizations and English-speaking countries. However, this material reflects the socio-cultural component of training that does not form the speaking skills aimed at solving specific legal issues via consultation

with the client, his support in the mediation process and in court proceedings. To change this we have to turn to those practices that are professionally oriented and efficient.

Legal education has had a long tradition of case method practice since the end of the nineteenth century. Its priority in the professional sphere is indicated by its implementation into the practice of special discipline study, such as social sciences, economics, medicine, chemistry, psychology, etc. [1–3]. Scientists distinguish the case method from other traditional teaching methods, pointing out its professionally-oriented nature as it develops the ability to apply theoretical knowledge to real-life cases [4–7], developing soft skills [8], self-regulation and teamwork skills [9], critical thinking [10,11] and conceptual understanding [12]. In terms of methodology, it allows realising the human-centred approach necessary for the graduates to succeed at work [13]. All of the above is reflected in the higher student's performance [14].

In recent years, higher education has been facing the challenges of distance learning as a measure compelled by COVID-19. All universities were forced to change the way of transmitting knowledge to students and receiving their feedback. The major part of offline courses were transformed into the distance format. Up to now the question of distance learning is disputable. The aim of this study is to compare the effectiveness of case technology in teaching ESP in virtual and face-to-face modes of instruction.

## 2. Literature Review

Starting in the 1990s, the case method has been used in teaching foreign languages among university students [15–17]. In the last decade, the implementation of the case method into teaching foreign languages to students of non-linguistic specialties has been actualised. The scope of E.A. Samorodova, M.K. Ogorodov, I.G. Belyaeva and E.B. Savelyeva's interest was the impact of cases in teaching Legal French [18], B.B. Levin carried out research on the case method in teaching future teachers [19], and V.V. Samoilova and N.Yu. Moroz tested it in terms of the specialty "Advertising and public relations" [20], etc. The introduction of the case method into the methodology of teaching foreign languages is currently increasing.

In parallel with interest in the pure case method, methodologists have started searching for ways of implementing it that could show high and permanent student results in the discipline. This predetermined the use of the method in the context of educational technology. The concept of *case technology* has appeared in the methodology of teaching foreign languages, though its nature is complicated by the existence of similar terms, such as *case method* [21,22], *case study* [15,23], *problem situation method* [24,25], etc. The range of terms relating to the concept of *case technology* raises the question of their synonymy.

The main issue is connected with the differentiation of two terms: *case method* and *problem situation method*. Despite their general similarity based on the problematic nature of the situations studied, they are not synonymous. A differential feature of these methods is the way of representing the case problem [26]: the case method actualizes the analytical activity, finding case difficulties from introductory information blocks, and the problem situation method introduces those difficulties in the open. The simplified mode of working on the case in the problem situation method limits the realization of the student's potential, which can help them restore the complete picture of the situation studied. On the contrary, the analysis of the initial circumstances of the case allows one to predict the possible risks of both participants in the process and identify the most effective ways to solve the problem. The actual difference between these two methods is significant when constructing a case in the context of a legal profile.

Another important issue is the differentiation of the terms *technology* and *method* in methodology. Methodologists view this issue in two aspects. The first one is connected with the content of these terms [27–29]. A *method* is defined as a group of techniques that ensure the implementation of a specific pedagogical task, whereas a *technology* is an algorithm that combines, as a rule, two or more interrelated methods [30,31].

The scope of methods that form case technology can be seen as a circle, where the case method is its core. The case method as an activity-based method that correlates with related methods aimed at activating the student's cognitive activity. The co-methods located in the periphery of the circle are the methods of an incident, business conference analysis method, situation analysis method, role-play, game modelling and the method of discussion (Figure 1) [32]. The implementation of these co-methods into the professional educational process allows improvement of the pure case method potential. Together with the methods aimed at solving problems and motivating students to foreign language learning, case technology includes methods used to assess the intermediate and final students' performance, i.e., a peer review method. The peer review method is practiced in the speaking skills formation process as a way of attaining the primary educational objective [33–35]. Nonetheless, it can also be used to solve more global educational tasks aimed at the formation of a comprehensively developed specialist, such as a lawyer. Thus, case technology is an algorithmized educational process that unites a number of activity-oriented methods based on the case method. Its implementation provides sustainable students' performance.

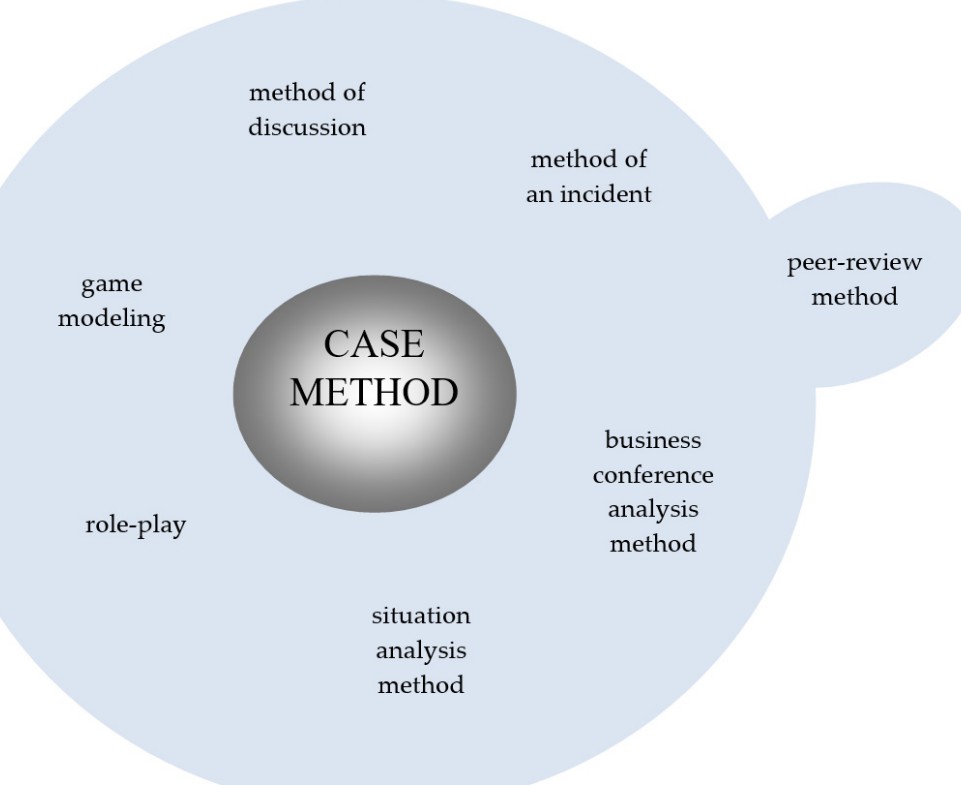

**Figure 1.** Case technology structure.

The second aspect of two term differentiation is determined by the implementation process. The case method implies the following sequence of actions: the study of general information on the case, the collection of additional information, decision-oriented thinking, comparison of all possible solutions, selection and justification of the most optimal solution. Whereas case technology may be introduced by several common stages that form the algorithm (their number varies in accordance with the case studied):

1.  situation introduction;
2.  analysis of the case problem;
3.  discussion of the case problem;
4.  presentation of the final solution to other groups of students (mini-groups);

5.  cross-discussion;
6.  assessment (peer review);
7.  summing up.

The efficiency of case technology based on the developed algorithm is one of its aspects as it is also connected with the way it is implemented into the educational process.

The COVID-19 pandemic forced universities to shift to online instruction. This shift caused the necessity of looking for new methods of implementing case technology in the virtual format of ESP teaching. Some researchers confirm the efficiency of distance learning on the whole and of some means used in this format in particular [36,37], though a group of researchers has pointed out some difficulties that both teachers and students have while studying online [38].

The analysis of the studies on distance learning show that the students' results usually vary and depend on different circumstances [39,40]. The performance of the course is of great importance for students and teachers. For students, the distance course has to be valuable, i.e., necessary in their future practice, reasonably informative, i.e., its content has to be relevant to the knowledge needed, and attractive, i.e., both about the clear and logical content outline and the 'appearance' of the course. For teachers, the performance is closely connected with methodology adapted to distance learning and the way the course is administered.

The teacher has to use technologies that consist of methods relevant to offline mode and the ones that are directed towards self-studying as a key component of any distance course. At this point, a lot depends on the student's personality, on the abilities required in the particular distance course and on their intentions and motives. Consequently, the priority issue raised is the student's motivation as a combination of external factors that show the attractiveness of the course to the student and their personal attitude to the discipline and learning on the whole. Student's motivation is determined by the teacher's support throughout the course. Some recent studies register the direct dependence of student's motivation on their progress: the less the teacher's involvement in the educational process is, the lower the student's results are [41–43]. Teacher's support is regarded as a guiding and systematizing component of training; the teacher clarifies tasks by depicting the challenges the student has to face, directs them when deviating from the general direction of the course and encourages them if necessary, etc.

The student's motivation is accompanied by their self-regulation—the ability to plan their work and meet the course deadlines. Distance courses are based on a flexible schedule that is, in the students' opinion, the unconditional benefit of the offline mode. However, the questionnaires show that students change their attitude to distance learning as they find out that, firstly, they do not have time-management skills formed by the time the course starts and, secondly, they have no proper motivation to start and complete the course [38].

In this research, the authors use the professionally-oriented case technology aimed at developing law students' speaking skills in the juridical context. The effectiveness of case technology in offline teaching mode, as proven by Almazova et al. [33], has to be verified in the course of distance learning, which is the aim of the research conducted. The research design was utilized to answer the following question: "How does online instruction influence the effectiveness of case technology in teaching ESP to law students?"

## 3. Materials and Methods

### 3.1. Research Design

An exploratory inductive approach was applied to this study, with the aim of determining the influence that distance learning has on the effectiveness of case technology in developing law students' English-speaking skills in the juridical context in the course of distance and face-to-face learning. The researchers employed mixed qualitative and quantitative methods to collect and analyse data for the study. Students performed their final assessment projects orally, based on the case technology. The qualitative observational method using Kim's scoring rubric [44] (see Appendix A) was applied to measure students'

performance, and the collected data consisted of the participants' scores for the presented projects. Subsequently, quantitative analysis of the final scores was carried out. The researchers used descriptive statistical analysis and comparative analysis of quantitative data, which provided them with new insights and detailed results.

*3.2. Participants*

The study took place at Peter the Great Saint Petersburg Polytechnic University, Russia. The study was carried out in the period from January 2020 to July 2021. Two groups of people were involved in collecting data for this study: 60 law students of the 2019–2020 academic year (Group A) and 60 law students of the 2020–2021 academic year (Group B). Participants were studying English for specific purposes (ESP) in the field of Law, which took place in the fourth term and lasted for one semester. Students of both groups were provided with the same syllabus content and studying materials during the ESP course. The participants' selection was based on the exam results of the English for general purposes (EGP) course, having been studied during the previous three terms of their education. The students in both groups had approximately the same level of English language proficiency at the beginning of the ESP course, the vantage or upper-intermediate level (B2), according to the Common European Framework of Reference for Languages (CEFR). Overall, 120 students were involved in the study, and all respondents volunteered to take part in the research.

*3.3. Data Collection*

The study was conducted in two stages. At the first stage (February–June 2019–2020), Group A students studied the ESP course face-to-face at a traditional campus setting. Researchers implemented the case technology as a method of teaching ESP to law students. During the term, students studied and practiced real cases related to various kinds of law (i.e. contract law, civil law, criminal law, etc.). Each case study lasted for two lessons and was devoted to a particular type of law. In the first lesson, students were introduced to the main notions and concepts relevant to a particular type of law, its types and characteristic features, terminology and precedent cases in this field of law, as well as different collocations, fixed expressions, phrases and vocabulary on the topic. In the second lesson, students were divided in groups of 6–8 people and given a real case and its background to analyse the situation and find the optimal solution to the case problem. Their group work involved the following stages:

1.  situation introduction;
2.  analysis of the case problem;
3.  discussion of the case problem;
4.  presentation of the final solution to other groups of students (mini-groups);
5.  cross-discussion;
6.  assessment (including peer review);
7.  summing up.

Each group performance consisted of three parts: legal advice to a client, mediation and court proceedings. Students' English oral communication included the combination of monological speech (i.e., opening and closing statements in court proceedings, court's judgment, etc.) and dialogical speech (i.e., initial consultation with a client, mediation, (witness) examination-in-chief, cross-examination, re-examination, etc.), as well as such forms of interaction as discussion and cross-discussion with other groups. Following that, students were set to prepare a case summary as part of their homework assignment, containing the case description, the process of solving the case problem and reaching the final solution and their attitude to the real court's judgment, which was revealed to students after the case discussion.

At the end of the term, students were to present their group projects as a form of final assessment for the course. Students were divided into groups of 6–8 people; each group chose a case from a variety of cases offered by a teacher and had a month to prepare

the project. Students were to analyse the case problem, consider and elaborate on the strategy and tactics to reach the desired outcome and present the final solution to the case problem. Students' projects included three parts: legal advice to a client, mediation and court proceedings. Groups presented their projects in the last lesson of the term; the reporting session was run according to the sequence of case technology stages given above. The researchers employed the observational method to collect data for this study. The researchers monitored students' performance and measured it using the assessment criteria of Kim's scoring rubric (see Appendix A). This scoring rubric has been validated in a study by Riaz, Sham and Riaz. [45]. Kim's rubric consists of 5 categories, i.e., meaningfulness, grammatical competence, discourse competence, task completion and intelligibility, in compliance with which the presented projects were scored. Students were given a particular score for each category, with the scales ranging from 0 to 5, where 0 means "No" control and 5 means an "Excellent" control over the given parameter. The assessment criteria (Kim's scoring rubric) had been given and explained to students at the beginning of the course.

During the second stage of the study (February–June 2020–2021) Group B students studied the ESP course having the same syllabus content and studying materials, although this time the educational process shifted to a distance learning format through an online teaching platform due to the COVID-19 pandemic and the introduction of quarantine measures. The researchers applied the case technology in teaching ESP to law students and followed the same procedure and practices as in the first stage of the study. Classes were arranged in MS team meeting rooms in the video conferencing format, where teachers and students could communicate and collaborate with each other. At the beginning of the term, the students received the criteria (Kim's scoring rubric) for the assessment of their oral communication and project performance, along with detailed explanations for them. During the term, the students studied real cases and applied their knowledge to real-life examples in different fields of law. Each case study lasted for two lessons and was devoted to a particular kind of law. In the first lesson, teachers introduced the main notions and concepts, terminology and precedent cases relevant to a particular kind of law, as well as different collocations, fixed expressions, phrases and vocabulary on the topic. In the second lesson, teachers divided students into groups of 6–8 people and introduced a real case and its background to analyse the situation and find the optimal solution to the case problem. The teachers organised students' group work in teams by creating a separate channel for each group where students could start their own meeting and discuss the case. Students analysed the case problem, shared their ideas on how to solve it and prepared for the presentation of the final solution to other groups of students. The teachers joined each group's meeting to monitor the group discussions and teamwork. All groups of students then returned to one common meeting and presented their final solution of the case problem to the class, followed by the subsequent cross-discussion of the solutions among the students, the assessment and summing up. At the end of the term, the students prepared and delivered their final assessment projects. Their performance and English-speaking skills were measured using Kim's scoring rubric to obtain data for the study.

## 4. Results

The researchers carried out the analysis of the data obtained at the first and second stage of the study using SPSS Version 23 (IBM Corp., Armonk, NY, USA, 2015 https://www.ibm.com/support/pages/downloading-ibm-spss-statistics-23, accessed on 8 July 2022). Firstly, the authors conducted descriptive statistical analysis of the data to identify the frequencies in participants' scores for each category of Kim's scoring rubric (see Table 1). A series of independent sample *t*-tests was then carried out to calculate the means and standard deviations for each scoring category and identify the relationship between these findings (see Table 2, Figure 2). The statistically significant difference between the results of two groups of students was proven by the *t*-test *p*-values (df-118, $\alpha = 0.05$).

**Table 1.** Participants' project grades according to Kim's scoring rubric in percentage.

| Assessment Criteria | Scales | | | | | | | |
|---|---|---|---|---|---|---|---|---|
| | Excellent | | Good | | Adequate | | Fair | |
| | Group A 2019–20 | Group B 2020–21 | Group A 2019–20 | Group B 2020–21 | Group A 2019–20 | Group B 2020–21 | Group A 2019–20 | Group B 2020–21 |
| Meaningfulness | 23.3% | 10% | 63.3% | 31.7% | 13.3% | 53.3% | 0% | 5% |
| Grammatical competence | 13.3% | 8.3% | 61.7% | 58.3% | 25% | 33.3% | 0% | 0% |
| Discourse competence | 28.3% | 8.3% | 58.3% | 48.3% | 13.3% | 38.3% | 0% | 5% |
| Task completion | 33.3% | 11.7% | 56.7% | 41.7% | 10% | 46.7% | 0% | 0% |
| Intelligibility | 13.3% | 8.3% | 60% | 61.7% | 26.7% | 30% | 0% | 0% |

**Table 2.** Mean score of the students' performance according to Kim's scoring rubric.

| Assessment Criteria | Results (Means and Standard Deviations) | | |
|---|---|---|---|
| | Group A (2019–2020 Academic Year) | Group B (2020–2021 Academic Year) | *t*-test Sig. (2-Tailed) df—118 $\alpha = 0.05$ |
| Meaningfulness | 4.10 SD—0.602 | 3.47 SD—0.747 | 0.000 |
| Grammatical competence | 3.88 SD—0.613 | 3.75 SD—0.600 | 0.231 |
| Discourse competence | 4.15 SD—0.633 | 3.60 SD—0.718 | 0.000 |
| Task completion | 4.23 SD—0.621 | 3.65 SD—0.685 | 0.000 |
| Intelligibility | 3.87 SD—0.623 | 3.78 SD—0.585 | 0.452 |

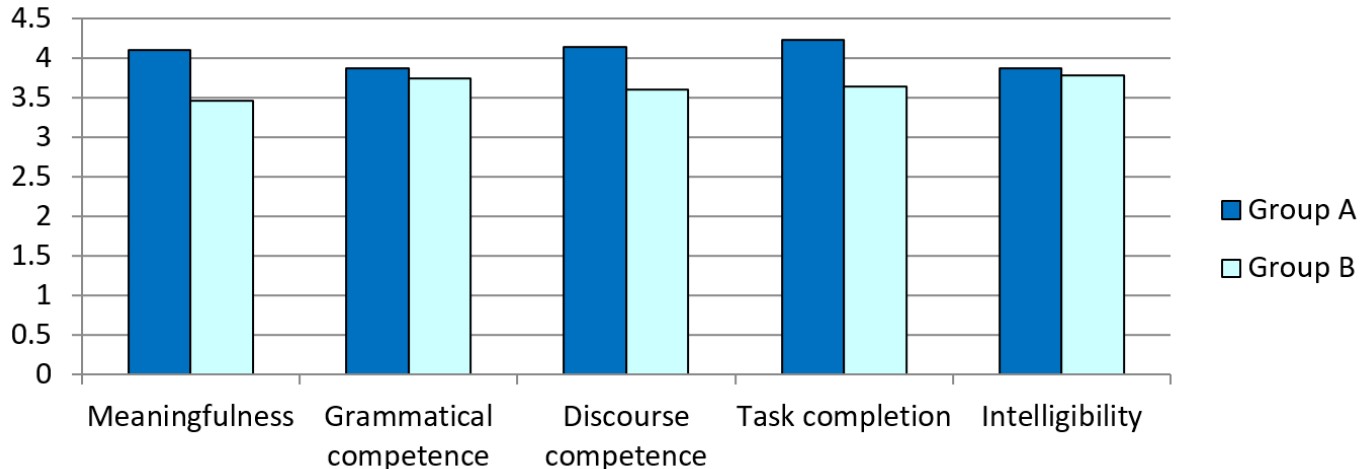

**Figure 2.** Mean score of the students' performance according to Kim's scoring rubric.

The obtained results showed no significant difference in such assessment criteria as "Grammatical competence" (3.88 and 3.75 for groups A and B, respectively) and "Intelligibility" (3.87 and 3.78 for groups A and B, respectively). About two thirds of students, from both groups, A and B, demonstrated a good level of grammatical competence: their responses were generally grammatically accurate without any major errors (e.g., article usage, subject/verb agreement, etc.) that could obscure the meaning. A similar pattern can be observed in the second criteria—Intelligibility. Although the responses of the major proportion of students (60% and 61.7% for groups A and B, respectively) included minor

difficulties with pronunciation or intonation, they were generally intelligible, clear, fluid and sustained, and did not require much effort by the listener.

The analysis of the students' speaking performance conducted in the academic years 2019–2020 (Group A) and 2020–2021 (Group B) and comparison of the obtained results provide us with certain ideas on the influence of distance learning on the effectiveness of case technology in the process of ESP acquisition. The quantitative analysis of the data allowed us to establish that, for such assessment criteria as "task completion", the percentage of students who achieved the highest score in group A is three times as much as in group B; 33.3% of the assessed students from group A received an excellent mark, whereas there were only 11.7% in group B. A satisfactory mark was given to 10% of students from group A and 46.7% of students in group B. Although all the students managed to meet the requirements for this criterion, the mean score of the students' performance according to Kim's scoring rubric is 4.23 and 3.65 for groups A and B, respectively, which demonstrates a better performance of students who studied in a traditional form. They displayed a more accurate understanding of the prompts without any misunderstood points in comparison with the representatives of Group B. Most students successfully covered all the main points with the complete details discussed in the prompts.

A similar pattern was seen in the results concerning other criteria. One more considerable difference between the two analysed groups was noted in the "discourse competence" criterion, which means that student's responses in Group A were more coherent and logically structured; these included the logical development of ideas and logical openings and closures. Students of Group A also demonstrated a better connection and transition of ideas by means of various cohesive devices (logical connectors, a controlling theme, repetition of key words, etc.) than their counterparts from Group B. As can be seen from Tables 1 and 2, the average score in Group A and B was 4.15 and 3.60, respectively. The number of students of Group A who showed excellent results is significantly higher (28.3%) than in Group B. Moreover, a relatively small proportion of students from Group B (5%) did not manage to completely meet the requirements. Their responses were loosely organized, which resulted in generally disjointed discourse and contained parts that displayed illogical or unclear organization, causing some confusion.

The same trend can be observed in figures for such assessment criterion as "meaningfulness". The students of Group A demonstrated a higher level of meaningfulness (the conveyed ideas were clearer and easier to understand) in comparison with the results of the students of Group B: 4.10 and 3.47, respectively. As for the excellent grade, there were twice as many students in Group A compared to Group B—23.3% and 10%, respectively. Similarly, the percentage of students who achieved good results is twice as much in Group A as in Group B (4.10 and 3.47, respectively).

Overall, the results show that, according to Kim's scoring rubric, the performance of students who studied in traditional face-to-face learning format was better than that of students who studied on-line. It can be concluded that the use of case technology in face-to-face format was more effective for the needs of improvement of students' English-speaking skills than the one carried out on-line.

## 5. Discussion

The Pandemic caused universities to shift quickly from in-person teaching to online teaching. The aim of this study was to investigate whether the online format was as effective as the traditional one. The results demonstrated that the distance learning format affected some of the assessed criteria of the final case presentation.

We did not observe any changes in the assessment criteria "Grammatical competence" and "Intelligibility", which was generally expected, due to the fact that initially the selected groups had approximately the same level of language proficiency. During the implementation of the projects, no additional effort was made to develop pronunciation skills and to master grammar; in this regard, no significant differences were found.

Because the progress of work on the project implied such steps as group discussion, presentation of the final solution to other groups of students (mini-groups), cross-discussion, assessment (peer review), etc., various communication options were implied. Students' English oral communication included the combination of monological and dialogical speech as well as such forms of interaction as a discussion and a cross-discussion with other groups. Undoubtedly, despite the high level of digitalization of the educational process and a wide range of technologies used, communication in the distance format was not organized as efficiently as in the traditional format. On-line learning was performed via various digital means of organizing communication of participants in the educational process (video conferencing, messengers, communication tools integrated into LMS), though it was difficult for the teachers to monitor and control the process of discussion within the groups. When working in small groups, students worked on separate channels, and the teacher could not track the work of the entire class. Consistently checking the work of individual groups, the teacher had no opportunity to assess the involvement of each of the participants in the discussion process. Moreover, during the discussion, students often switched to their native language or did not participate in the discussion due to the fact that they were distracted by extraneous activities. This supports the findings that the results of distance learning are not permanent [39,40] and depend on the student's personality and acquired abilities. For example, the distance format of learning can impede academic performance if students lack motivation or have an insufficient level of self-organisation and metacognition [41,42].

This study revealed many positive outcomes related to the use of case technology in teaching professional law communication to law students, which supports the findings of [4–12,18–22,33]. One of the key findings is that students who practiced the case technology in a traditional format completed the task more successfully than those who studied on-line; they covered all the main points with complete details discussed in the prompt, displayed smooth connection and transition of ideas by means of various cohesive devices (logical connectors, a controlling theme, repetition of key words, etc.), their responses were generally meaningful and the conveyed ideas were clearer and easier to understand. These results support the conclusions made by Ya and Katz [46,47] that student performance is mostly independent on the mode of instruction, providing it contains several components: the appropriate content of the online learning, the features and the ease of use of the learning management platform, as well as the interactivity between the students and the lecturer, and among the students themselves. Obviously, the on-line format of the case technology did not manage to meet these requirements, which resulted in worse results of students' performance. Because it was impossible to achieve the same level of interactivity among the students, those who studied on-line did not have as much speaking practice as those who took a course in a traditional format.

The use of the traditional form of case technology showed more expediency during the experiment. The consistency of the results and better performance of Group A must be treated as evidence that proper ESP teaching material development and its adaptation to the needs of distance learning is necessary when practicing case technology on-line and is vital for the improvement of student's English-speaking skills. This supports the ideas found in numerous studies [38–43].

The findings of [43,48–50] demonstrate that the effective construction of the ESP teaching system requires such aspects as teaching materials, teaching methods, teacher training and evaluation. The creation of personalized teaching materials and multiple activities can help students to improve their English-speaking skills [51,52]. In this regard, for the elimination of the negative consequences of the distance learning on the use of case technology, it is necessary to develop recommendations and instructions for on-line case technology in ESP courses.

Moreover, the motivational component of distance learning is not disputable, as the lack or low level of support minimizes the students' activity [42,43]. A few students can meet the requirements and show perseverance, self-control and initiative, which is why

motivational strategies are regarded as the obligatory element of distance learning [48]. Due to this, we recommend developing some tasks to increase students' motivation for learning activities on-line, in particular for promoting group-discussions and better interaction within mini-groups.

As it has already been mentioned, students often lack the ability to self-regulate due to insufficiently-developed metacognitive skills [41,53]; teaching students to plan independent academic work in order to increase the efficiency of mastering the material may improve the way students complete the tasks and lead to higher academic achievements.

Despite being a popular topic for investigation [5,10,18,33], very little is known regarding the outcomes associated with the case technology performed in the distance learning format. This study contributes to the issue of on-line ESP instruction. The given research provides another example of how the distance learning and face-to face formats of ESP instruction differ, and what constraints distance learning can apply to the case technology in teaching ESP. The achieved results make it possible to conclude that the traditional face-to-face form of case technology was more efficient for the improvement of English-speaking skills. We consider that a proper design of ESP teaching materials adapted to the needs of distance learning may allow students to develop a productive mechanism for improving monological speaking skills in the process of practicing case-study technology.

**Author Contributions:** Conceptualization, O.S. and T.K.; data curation, I.K. and L.V.; formal analysis, O.S.; investigation, O.S., I.K., L.V. and T.K.; methodology, O.S., I.K. and L.V.; project administration, T.K.; resources, I.K.; software, O.S., I.K. and L.V.; supervision, T.K.; validation, O.S., I.K. and L.V.; visualization, L.V.; writing—original draft, O.S., I.K., L.V. and T.K.; writing—review and editing, O.S., I.K. and L.V. All authors have read and agreed to the published version of the manuscript.

**Funding:** The research was funded by the Ministry of Science and Higher Education of the Russian Federation under the strategic academic leadership program 'Priority 2030' (Agreement 075-15-2021-1333 dated 30 September 2021).

**Institutional Review Board Statement:** The study was conducted in accordance with the Declaration of Helsinki and approved by the Institutional Review Board of the Institute of Humanities, Peter the Great St. Petersburg Polytechnic University (protocol code 5, dated 31 January 2020).

**Informed Consent Statement:** Informed consent was obtained from all subjects involved in the study.

**Data Availability Statement:** Data reported are available upon reasonable request from the corresponding author.

**Acknowledgments:** The authors express their gratitude to the reviewers who carried out the analysis and constructive criticism of the submitted article, as well as to all participants of the experiment. The authors are grateful to the organizers of the conference "Professional Culture of the Specialist of the Future" (Peter the Great St. Petersburg Polytechnic University).

**Conflicts of Interest:** The authors declare no conflict of interest.

## Appendix A

**Table A1.** Kim's (2010) Analytic Scoring Rubric.

| Analytic Scoring Rubric | |
| --- | --- |
| **Meaningfulness** | (Communication Effectiveness) Is the response meaningful and effectively communicated? |
| **Grammatical Competence** | Accuracy, Complexity and Range |
| **Discourse Competence** | Organization and Cohesion |
| **Task Completion** | To what extent does the speaker complete the task? |
| **Intelligibility** | Pronunciation and prosodic features (intonation, rhythm, and pacing) |

**Table A1.** *Cont.*

| Meaningfulness (Communication Effectiveness) Is the response meaningful and effectively communicated? | | | | | |
|---|---|---|---|---|---|
| **5 Excellent** | **4 Good** | **3 Adequate** | **2 Fair** | **1 Limited** | **0 No** |
| The response is completely meaningful—what the speaker wants to convey is completely clear and easy to understand. It is fully elaborated and delivers sophisticated ideas. | The response is generally meaningful—in general, what the speaker wants to convey is clear and easy to understand. It is well elaborated and delivers generally sophisticated ideas. | The response occasionally displays obscure points; however, the main points are still conveyed. It includes some elaboration and delivers somewhat simple ideas. | The response often displays obscure points, leaving the listener confused. It includes little elaboration and delivers simple ideas. | The response is generally unclear and extremely hard to understand. It is not well elaborated and delivers extremely simple, limited ideas. | The response is incomprehensible. It does not contain enough evidence to evaluate. |

| Grammatical Competence: Accuracy, Complexity and Range | | | | | |
|---|---|---|---|---|---|
| **5 Excellent** | **4 Good** | **3 Adequate** | **2 Fair** | **1 Limited** | **0 No** |
| The response is grammatically accurate. It displays a wide range of syntactic structures and lexical forms. It displays complex syntactic structures (relative clause, embedded clause, passive voice, etc.) and lexical forms. | The response is generally grammatically accurate without any major errors (e.g., article usage, subject/verb agreement, etc.) that obscure meaning. It displays a relatively wide range of syntactic structures and lexical forms. It displays relatively complex syntactic structures and lexical forms. | The response rarely displays major errors that obscure meaning and a few minor errors (but what the speaker wants to say can be understood). It displays a somewhat narrow range of syntactic structures; too many simple sentences. It displays somewhat simple syntactic structures. It displays the use of somewhat simple or inaccurate lexical forms. | The response displays several major errors as well as frequent minor errors, sometimes causing confusion. It displays a narrow range of syntactic structures, limited to simple sentences. It displays the use of simple and inaccurate lexical forms. | The response is almost always grammatically inaccurate, which causes difficulty in understanding what the speaker wants to say. It displays lack of basic sentence structure knowledge. It displays generally basic lexical forms. | The response displays no grammatical control. It displays severely limited or no range and sophistication of grammatical structure and lexical form. It does not contain enough evidence to evaluate. |

| Discourse Competence: Organization and Coherence | | | | | |
|---|---|---|---|---|---|
| **5 Excellent** | **4 Good** | **3 Adequate** | **2 Fair** | **1 Limited** | **0 No** |
| The response is completely coherent. It is logically structured—logical openings and closures and logical development of ideas. It displays smooth connection and transition of ideas by means of various cohesive devices (logical connectors, a controlling theme, repetition of key words, etc.). | The response is generally coherent. It displays a generally logical structure. It displays good use of cohesive devices that generally connect ideas smoothly. | The response is occasionally incoherent. It contains parts that display somewhat illogical or unclear organization; however, as a whole, it is in general logically structured. It at times displays a somewhat loose connection of ideas. It displays the use of simple cohesive devices. | The response is loosely organized, resulting in generally disjointed discourse. It often displays illogical or unclear organization, causing some confusion. It displays repetitive use of simple cohesive devices; use of cohesive devices are not always effective. | The response is generally incoherent. It displays illogical or unclear organization, causing great confusion. It displays attempts to use cohesive devices, but they are either quite mechanical or inaccurate, leaving the listener confused. | The response is incoherent. It displays virtually non-existent organization. It does not contain enough evidence to evaluate. |

**Table A1.** *Cont.*

| Task Completion To what extent does the speaker complete the task? | | | | | |
|---|---|---|---|---|---|
| **5 Excellent** | **4 Good** | **3 Adequate** | **2 Fair** | **1 Limited** | **0 No** |
| The response fully addresses the task and displays completely accurate understanding of the prompt without any misunderstood points. It completely covers all main points with complete details discussed in the prompt. | The response addresses the task well and Includes no noticeably misunderstood points. It completely covers all main points with a good amount of details discussed in the prompt. | The response adequately addresses the task and includes minor misunderstandings that do not interfere with task fulfillment. It touches upon all main points, but leaves out details **OR** completely covers one (or two) main points with details, but leaves the rest out. | The response insufficiently addresses the task and displays some major incomprehension/ misunderstanding(s) that interferes with addressing the task **OR** touches upon bits and pieces of the prompts. | The response barely addresses the task and displays major incomprehension/ misunderstanding(s) that interferes with addressing the task. | The response shows no understanding of the prompt. It does not contain enough evidence to evaluate. |
| Intelligibility Pronunciation and prosodic features (intonation, rhythm and pacing) | | | | | |
| **5 Excellent** | **4 Good** | **3 Adequate** | **2 Fair** | **1 Limited** | **0 No** |
| The response is completely intelligible although accent may be there. It is almost always clear, fluid and sustained. It does not require listener effort. | The response may include minor difficulties with pronunciation or intonation, but generally intelligible. It is generally clear, fluid and sustained. Pace may vary at times. It does not require listener effort much. | The response may lack intelligibility in places impeding communication and exhibits some difficulties with pronunciation, intonation or pacing. It exhibits some fluidity. It may require some listener efforts at times. | The response often lacks intelligibility impeding communication and frequently exhibits problems with pronunciation, intonation or pacing. It may not be sustained at a consistent level throughout. It may require significant listener effort at times. | The response generally lacks intelligibility and is generally unclear, choppy, fragmented or telegraphic. It contains frequent pauses and hesitations, consistent pronunciation and intonation problems. It requires considerable listener effort. | The response completely lacks intelligibility. It does not contain enough evidence to evaluate. |

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
