# Peer review of "Case Technology in Teaching Professional Foreign Communication to Law Students: Comparative Analysis of Distance and Face-to-Face Learning"

_education, doi:10.3390/educsci12100645_

Round 1
Reviewer 1 Report
1. “Case-study technology” and “Case technology” are both mentioned in the abstract, but “Case technology” in the title. This is rather confusing, please explain further about this issue.
2. Longitudinal study was also mentioned, is this a case study or not? Case studies is a way of data collection while the longitudinal studies deal with time for the data collection or it can be a way to measure one thing over and over again. Which of these two methods is used in this study?
3. Further, neither of the above two research methods are quantitative study, I wonder why the author(s) stated that “Quantitative methods were applied to collect and analyse data for the study”. If as the paper presented, case method is a method other than the above-mentioned methods, then case method needs to be elaborated in detail.
4. However, in the Introduction and Literature Review, the author(s) stated that case method practice is a tradition of legal education, and several terms has mentioned, such as case method, case study, problem situation method, case technology, but did not explain how these concepts can be used to teach professional foreign communication to law students.
5. No reference is cited to the case method structure presented in Figure 1. The author(s) will have to explain what the theoretical background of this structure is.
6. In the Materials and Methods, the author(s) claimed that the two group of students studied the same materials in the same procedure, but failed to explain the detail of how they can do this in face to face and online environment. There are some certain procedures that will never be the same in these two different learning environments.
7. The results were based on the evaluation of students’ performance of the final assessment which was measured by Kim’s scoring rubric. Giving that the validity and radiality of the assessment and the rubric were not provided, the results were questionable.
8. There are references in languages other than English which need to be translated into English, e.g. 16, 20.
9. Some of the references do not seem to fit into the scope of this study, such as 4, 5, 6, 22, etc.
Author Response
We appreciate the time and effort that you dedicated to providing feedback on our manuscript and are grateful for the insightful comments on and valuable improvements to our paper. We have incorporated most of the suggestions you have made. Please see the attachment.

Reviewer 2 Report
The paper discusses a quite compelling issue of teaching online compared to face-to-face formats within ESP lessons. In the theoretical part the key terms are precisely introduced and described. The research part is clear and understandable, however, from the point of view of a form, I have several suggestions of how to correct the paper:
Please explain Kim´s scoring rubric (I have not found an explanation, nor reference for this type of assessment),
There is a mistake in the citaton (a reference is used in the text), line 235, and 236, please correct.
Line 257 – use of comma instead of point in the numerical result, the same for 269, 282 and 290.
Please, unify the numerical data that is used in two forms in the text.
Author Response

(The authors gave the same response as above.)

Reviewer 3 Report
If the introduction and literature review parts are given separately, the problem situation will be more understandable.
In order to make the aim of the study more understandable, research questions can be added and the findings can be arranged in parallel with these questions.
The method of the study should be better structured. It can be presented under the subheadings of research design, participants, data collection tools, procedure and data analysis.
The contribution of the study to the international literature should be discussed in more detail.
The results obtained in the Discussion section should be compared with the literature.
Author Response

(The authors gave the same response as above.)

Round 2
Reviewer 1 Report
After reviewing the revised manuscript, I feel that the manuscript is substantially improved after making the suggested edits. It can be accepted in its present form now.